# Do Human Milk Oligosaccharides Protect Against Infant Atopic Disorders and Food Allergy?

**DOI:** 10.3390/nu12103212

**Published:** 2020-10-21

**Authors:** Soo Min Han, Aristea Binia, Keith M. Godfrey, Sarah El-Heis, Wayne S. Cutfield

**Affiliations:** 1Liggins Institute, The University of Auckland, Auckland 1023, New Zealand; clara.han@auckland.ac.nz; 2Nestlé Research, Société des Produits Nestlé SA, 1000 Lausanne, Switzerland; Aristea.Binia@rdls.nestle.com; 3NIHR Southampton Biomedical Research Centre, University of Southampton and University Hospital Southampton NHS Foundation Trust, Southampton SO17 1BJ, UK; kmg@mrc.soton.ac.uk; 4MRC Lifecourse Epidemiology Unit, University of Southampton, Southampton SO17 1BJ, UK; se@mrc.soton.ac.uk; 5A Better Start—National Science Challenge, Liggins Institute, The University of Auckland, Auckland 1023, New Zealand

**Keywords:** atopic disorders, breastfeeding, food allergy, human milk oligosaccharides

## Abstract

Atopic disorders (AD), often coexistent with food allergy (FA), start developing in early life and have lifelong health consequences. Breastfeeding is thought to be protective against AD and FA, but the data are controversial, and mechanisms are not well understood. Human milk oligosaccharides (HMOs) are complex carbohydrates that are abundant in human milk. These are thought to contribute to the development of the infant immune system by (i) promoting healthy microbiome, (ii) inhibiting pathogen binding to gut mucosa and (iii) modulating the immune system. Differences in microbiome composition between allergic and healthy infants have been observed, regardless of breastfeeding history. To date, limited studies have examined the preventive effects of HMOs on AD and FA in infants and current data relies on observation studies as trials of varying HMO intake through randomising individuals to breastfeeding are unethical. There is evidence for beneficial effects of breastfeeding on lowering the risks of FA, eczema and asthma but there are inconsistencies amongst studies in the duration of breastfeeding, diagnostic criteria for AD and the age at which the outcome was assessed. Furthermore, current analytical methods primarily used today only allow detection of 16–20 major HMOs while more than 100 types have been identified. More large-scale longitudinal studies are required to investigate the role of HMO composition and the impact of changes over the lactation period in preventing AD and FA later in life.

## 1. Atopic Disorders and Food Allergy in Childhood

Atopy is an exaggerated immune reaction leading to production of immunoglobulin E (IgE) antibodies in response to external stimulus [1]. The most common atopic disorders (AD) in children are eczema, asthma and allergic rhinitis. Eczema is characterised by red, scaly and itchy skin [2], asthma by recurrent wheezing [3] and allergic rhinitis by nasal congestion [4]. Food allergy (FA) also starts developing during early life which is an adverse immunological (hypersensitivity) response to food [5]. Food sensitization refers to the presence of allergic antibodies which may not lead to overt allergic reaction. Food sensitization precedes FA in some children, but it does not always develop into FA later in life [6]. Symptoms of FA range from urticaria to anaphylaxis, in severe cases [7]. Diagnosis of FA requires invasive tests such as skin prick testing, serum-specified IgE testing to foods, or oral food challenges.

Approximately 20% of children have eczema and 10% have asthma across the globe [1] and the childhood prevalence of AD and FA are increasing worldwide. In 2011, a US survey revealed that 8% of children under 18 years old have FA. [8]. In children, the most common types of food allergies are cow’s milk, egg, peanut, fish and tree nut allergies [7]. Cow’s milk and egg allergies often disappear by adolescence whereas peanut, fish and tree nut allergies tend to persist throughout adulthood [1,9]. In the US, the prevalence of eczema in the same age group increased from 7.4% in 1997–1999 to 12.5% in 2009–2011 [10]. In New Zealand, 18% of children aged 0–4 years were diagnosed with eczema in 2018–2019, compared to 14.8% in 2006–2007 [11].

There is a genetic pre-disposition for AD and FA, with greater risks in children with two atopic parents [1]. Skin barrier impairment is a major trigger for eczema, often associated with a filaggrin gene variant [12]. This is often the first manifestation of AD and further increases the risk of FA [13]. For example, it was discovered that filaggrin gene variants were also associated with peanut allergy [14]. A healthy immune system development may be a key modulator of AD and FA susceptibility. Adequate exposure to environmental microbes and allergens is one way to stimulate immunological development [15]. Emerging evidence points to immunomodulatory functions of the gut microbiome [16]. During early life, delivery mode, feeding practices, antibiotic exposure and probiotic supplements can affect microbiome shaping and composition [17], which may have a significant impact on allergy risks in later life. Management of AD or FA into adulthood may require avoidance of certain foods or life-long medication use such as corticosteroids, anti-histamines or bronchodilators, potentially greatly impacting the quality of daily life [18]. Early prevention or modulation of AD is therefore essential.

## 2. Does Breastfeeding Modulate AD and FA Risks?

The World Health Organisation currently recommends infants to be exclusively breastfed for the first six months of life [19]. There are data to suggest beneficial outcomes of breastfeeding [20], including lower risks of obesity [21], type 2 diabetes [22], and pneumonia [23] as well as better cognitive function [24]. However, the PROmotion of Breastfeeding Intervention Trial (PROBIT) found that promotion of increased breastfeeding duration and exclusivity was beneficial for later verbal function but not for other neurocognitive domains or obesity [25,26].

There are conflicting findings on breastfeeding lowering AD and FA outcomes in infants. Any type of breastfeeding for ≥6 months and exclusive breastfeeding for ≥4 months was associated with decreased eczema risks at one and two years of age [27,28,29]. Specifically, Chiu et al. [27] showed that the protective effect on eczema was stronger with exclusive breastfeeding compared with partial breastfeeding. However, it is unclear if breastfeeding impacts eczema outcomes during childhood and adolescence. Exclusive breastfeeding for ≥4 months and ≥6 months was associated with lower prevalence of eczema at ages 10 [30] and 17 years [31], respectively. In contrast, Bergmann et al. showed that in the first seven years, the risk of eczema increased each year with each additional month of any type of breastfeeding in infants with a history of parental eczema [32]. Such contrasting findings may suggest that breastfeeding is not the best preventive mechanism against eczema in children with a genetic risk.

The risk of wheezing among infants was reduced by 62% with exclusive breastfeeding for ≥6 months [33]. In addition, there was a reported 33% reduced rate of wheezing in infants exclusively breastfed for ≥12 months compared to those breastfed for ≤6 months [33], suggesting a dose-dependent association. However, in infants breastfed by mothers with asthma, the rates of recurrent wheeze were reduced with ≥4 months of exclusive breastfeeding only up to and including three years of age [34]. The long-term effects of breastfeeding on asthma risks are less evident. In a New Zealand population, any breastfeeding for ≥4 weeks was associated with increased risks of asthma at nine years of age [35]. Other studies found that the risks decreased with breastfeeding at six years [36] but there was no association at 10 years of age [30]. The differential findings may be due to the different definitions used for asthma. Sears et al. [35] have included the diagnosis of hay fever in the definition while Elbert et al. [30] and Silvers et al. [36] have not. This may explain the increased prevalence of asthma observed in New Zealand children at nine years of age.

While breastfeeding duration did not influence FA development in infants by 1–3 years of age, early weaning and early introduction of solid foods (≤16 weeks) were the risk factors for FA [37,38]. Similarly, exclusive breastfeeding for ≥4 months [39], later introduction of formula milk (after six months of age) and solid foods (after one year of age) were associated with reduced FA risks in childhood [40]. However, a more recent study found that early introduction of peanut protein decreased the prevalence of peanut allergy in high-risk infants [41]. Hence, the timing of food allergen introduction may be a key modifier of FA risk in infants. If exclusive breastfeeding is prolonged, introduction of solid foods may be delayed, creating conflict between the two factors that influence FA outcomes. The current recommendation is that complementary foods can be introduced between four and six months of age and there is no strong evidence to support reduced FA risks with delayed introduction of solid foods [42].

Overall, meta-analyses have suggested that breastfeeding is protective against AD outcomes during infancy [43,44,45]. These associations were stronger in infants with a family history of AD [29,33]. However, current evidence is limited to observational studies as it is unethical to randomise infant feeding practices. Thus, unrecognised confounding factors could influence the findings of these studies. Different studies have used different durations of breastfeeding, diagnostic criteria of AD and follow-up periods [43]. These create great variations in outcomes, potentially causing over- or under- estimation of effects of breastfeeding on AD and FA. Moreover, findings are often confounded by the demographic background of the parents. Mother’s willingness to breastfeed exclusively was proportional to atopic risk level in the family and parents who received higher education were more likely to breastfeed for longer [28].

## 3. What is the Role of Human Milk Components in AD and FA Prevention?

AD and FA originate through immune reactions against external allergens [46]. A hyperresponsive immune system leads to an allergic inflammation cascade, with type 2 helper T (Th2) cells producing inflammatory cytokines, such as interleukin (IL)-4, IL-13, and IL-5, ultimately leading to increase in IgE production by B cells [46]. During early life, it is thought that breast milk contributes to the development of a balanced immune system that can appropriately tolerate external allergens without an inflammatory response [47].

Breast milk not only consists of essential nutrients for infants [48], including fats, carbohydrate and protein, but also contains bioactive molecules with immunologic activities, such as secretory immunoglobulin A, cytokines, transforming growth factors (TGF) and lactoferrin [49], and microbiomes [50]. Colostrum, the first milk produced, contains higher concentrations of immunologic molecules compared to mature milk [51]. This suggests that the primary role of colostrum is to provide protection for the infant against environmental pathogens. The infant’s immune system can be regulated by changes in the mother’s physiological environment. For example, an intervention study has shown probiotic supplementation for six months in the mother from 2–5 weeks before delivery increased the levels of IgA (at week one and three months) and TGF-β1 (at three months) in their breast milk [52]. In a separate study, probiotic supplementation in the mother from 36-weeks gestation until birth and continuation in the infant from birth until 12 months of age was associated with lower rates of IgE-associated eczema in the infant during the second year of life [53]. These studies suggest probiotics have the potential to modulate the infant’s immune function through improving maternal microbial environment. However, other environmental changes, such as antibiotic use by the mother around the time of delivery, can reduce the richness and diversity of microbiome composition in breast milk [54]. Meta-analyses show that breastfeeding is effective in preventing respiratory tract infections and diarrhoea in infants [55,56]. Necrotising enterocolitis (NEC) is prevalent among preterm and very low birth weight infants [57] but the risk is significantly reduced in breastfed infants [58].

Human milk oligosaccharides (HMOs) are complex carbohydrates that are the third most abundant solid component of human milk [48]. The structural variations of HMOs have been reviewed previously [59]. HMO structures are more complex and diverse compared to other sources of oligosaccharides (Figure 1), such as bovine milk oligosaccharides (BMO), and plant-derived oligosaccharides, gallacto-oligosaccharides (GOS) and fructo-oligosaccharides (FOS), which are supplemented in infant formula [60]. Briefly, monosaccharide building block structures like fucose and sialic acid are attached to lactose, producing fucosylated and sialylated HMOs. For example, fucosyltransferase 2 (FUT2), expressed by the Secretor gene (Se), adds a fucose by α1-2-linkages, producing HMOs such as 2′-fucosyllactose (2′FL) and lacto-*N*-fucopentaose I (LNFP-I). Fucosyltransferase 3 (FUT3), expressed by the Lewis gene (Le), adds fucose by α1-3/4-linkages, producing LNFP-II, for example [61]. Based on the concentrations of 2′FL (combined with LNFP-I), as proxy for FUT2 activity, and LNFP-II, as proxy for FUT3 activity, mothers can be divided into four milk groups: Lewis positive Secretors (Se^+^Le^+^), Lewis positive Non-secretors (Se^−^Le^+^), Lewis negative Secretors (Se^+^Le^−^), and Lewis negative Non-secretors (Se^−^Le^−^) [62]. Each milk group has a distinctive HMO profile and the composition varies significantly [61,63].

HMO content changes over the course of lactation, decreasing from approximately 20–25 g/L in colostrum to approximately 10–15 g/L in mature milk [64]. Additionally, individual HMO concentrations change dynamically during this time [63]. Further research is required to better understand if these changes in HMOs during the lactation phase modulate the risk of allergic disease. Concentrations of dominant HMOs 2′FL and LNFP-I were found to be highest at the start of lactation and gradually decreased [63]. This could indicate that FUT2 activity is highest early in lactation and that it competes for substrates with other enzymes, like FUT3, later in lactation. This could point to regulatory mechanisms of enzymes involved in HMO synthesis. In addition, 3-fucosyllactose (3FL), disialyllacto-*N*-tetraose (DSLNT) and 3′-sialyllactose (3′SL) concentrations were higher in milk collected later in lactation [65], suggesting a role later in infancy for infant physiological needs and development. Other factors that affect HMO composition include maternal body mass index and mode of delivery [63,65].

## 4. Could HMOs be Related to AD and FA Prevention?

There are three mechanisms in which HMOs affect the immunity of newborns [66]. Early life is a critical stage for maturation of the gut microbiome which may have an impact on allergy risks in later life [16]. One of the primary roles of HMOs is to serve as prebiotics for the gut microbiota [67]. Ingested HMOs resist degradation in the small intestine and reach the colon, where they act as metabolic substrates for *Bifidobacterium* species. This increases the production of short chain fatty acids like propionate and butyrate, which have anti-inflammatory properties [68]. Children with higher level of butyrate or propionate were less likely to develop asthma and food allergy by one year old [69]. Breastfed infants have been reported to have higher numbers of *Bifidobacterium* compared to formula-fed infants [70,71] and healthy microbiota may be associated with reduced risks of AD and FA. He et al. [72] observed that, among groups of infants who were breastfed, healthy infants had a typical infant type microbiome with high levels of *Bifidobacterium bifidum*, whereas allergic infants had an adult type microbiome with high levels of *Bifidobacterium adolescentis*. In other studies, lower gut microbiome diversity at age one week [73], one month [74] and three months [75] were associated with eczema by age 18 and 24 months and with food sensitization by one year, respectively. These findings were independent of factors influencing the gut microbiome, such as caesarean birth, antibiotic use, probiotic supplement and exclusive breastfeeding. The microbiota of infants with food allergy by age one year were characterised by reduced presence of *Bacteroidetes, Proteobacteria* and *Actinobacteria* and enriched with *Firmicutes* [76]. Overall, these studies suggest there are links between gut microbiome development in early infancy and later AD and FA susceptibility. Further studies exploring the interactions between HMOs, early gut microbiome development and atopic susceptibility are required.

Secondly, HMOs inhibit the adhesion of pathogens to the intestinal epithelium. Structurally, they resemble cell surface receptors which the pathogens attach to [66]. It was observed that diarrhoea occurred less often in breastfed infants and α1,2-fucosylated HMOs were associated with protection against diarrhoea caused by campylobacter and caliciviruses [77]. Interestingly, genes such as FUT2 might also influence the expression of H antigen which may mediate the attachment of pathogens and contribute to associations between FUT2 genetic variants and reported infections in early life [78]. Furthermore, the primary cause of NEC in preterm infants is disrupted gut barrier resulting from decreased mucus secretion, such as Mucin 2 (MUC2) and trefoil factor-3 (TFF3), by intestinal goblet cells [79,80]. In rat pups with NEC, oral administration, at postnatal day five, of extracted HMOs from human milk samples attenuated the morphology of NEC [80]. This was mediated through direct modulation of goblet cell function by HMOs which led to upregulation of MUC2 expression. Similar effects were observed in human intestinal cell line LS174T, where treatment with HMOs stimulated the expression of MUC2 and TFF3 in these cells [80]. The effects of individual HMOs were also studied, the strongest effect observed with 3FL: concentrations of 10 mg/L enhancing the expression of MUC2 and TFF3, while that of 15 mg/L upregulated only TFF3 expression [81]. Such effects of 3FL persisted even after exposure of the cells to inflammatory cytokines, which demonstrates the function of 3FL in enhancing gut barrier under pro-inflammatory environment. Hence, HMOs have a capacity to improve the gut barrier function, as well as the immune function in infants [66,82]. Whether this further modulates allergy risks in infants is yet to be fully investigated.

Lastly, HMOs can directly modulate the immune system. It was shown in vitro that HMOs reduce the production of pro-inflammatory cytokines and increase the levels of cytokines involved in tissue repair and homeostasis [83]. Specifically, HMOs 3′-, 4- and 6′-galactosyllactoses from colostrum was responsible for modulating the immune response by attenuating IL-8 production in immature intestinal epithelial H4 cells after pro-inflammatory stimulation [84]. It is proposed that 1% of HMOs are absorbed into the systemic circulation [64]. This suggests that in vivo, HMOs may suppress the inflammatory reaction associated with atopy.

There are several studies that examined the associations between individual HMOs and AD or FA outcomes in infants. There was no significant difference in consumption of neutral HMOs in colostrum by infants who developed eczema by 18 months (*n* = 9) and by infants who did not (*n* = 11) [85]. Others observed that the milk of mothers with an infant with cow’s milk allergy contained lower levels of 6′-sialyllactose (6′SL), DSLNT, LNFP-I and LNFP-III [86]. Interestingly, mother’s Secretor status was associated with infants having delayed-onset cow’s milk allergy rather than having an immediate-type (IgE-mediated) one. Similarly, additional studies are required to examine he implications of mother’s milk group on FA and/or AD outcomes in infants. For example, DSLNT concentrations was found to be higher in Se^−^Le^+^ and Se^−^Le^−^ mothers, compared to other milk groups [63]. However, lower DSLNT concentrations were observed in milk from mothers of preterm infants that developed NEC [87]. In another study, individual HMOs were not, but combinations of specific HMOs were, associated with food sensitization by 1 year of age [88]; lower risk of food sensitization was associated with relatively higher concentrations of fucodisialyllacto-*N*-hexaose (FDSLNH), LNFP-II, lacto-*N*-neotetraose (LNnT), LNFP-I, sialyllacto-*N*-tetraose c (LSTc) and fucosyllacto-*N*-hexaose (FLNH), and with relatively lower concentrations of lacto-*N*-hexaose (LNH), lacto-*N*-tetrose (LNT), 2′FL and disialyllacto-*N*-hexaose (DSLNH). It is uncertain if HMOs have a long-term impact on AD or FA outcomes during childhood. Sprenger et al. found that FUT2-dependent HMOs reduced risk of eczema at two years, but not at five years, among infants born by caesarean section [89]. Existing evidence is inconsistent and based on studies with varied designs and AD definitions, and it is uncertain if HMOs have a long-term impact on AD and/or FA. Future studies need to address these limitations to clarify the role of HMOs in allergy prevention during infancy and childhood.

## 5. What are the Limitations to Overcome?

There are several limitations in clinical and technical aspects of current study methods. As outlined earlier, individually randomising HMO feeding practices would be unethical, so the evidence relies on observational studies. As commercial HMO production increases, these are likely to be added to infant formula, allowing randomised controlled trials in the future. So far, few studies have investigated HMOs in relation to allergy outcomes in breastfed infants. The majority of studies in the literature have focused on breastfeeding only and not on HMOs per se. While some studies have compared exclusive breastfeeding with formula feeding [28] or no breastfeeding [39], others have compared different breastfeeding durations (<4 months vs. ≥4 months [29]; six months vs. 12 months [33]), which may be the key factor. This might imply that the resulting associations are only applicable to specific group of infants but not to others. In addition, studies have used different follow-up periods and the age at which the infant is assessed may be important. First symptoms of AD and FA usually appear during infancy [1] and this is when the exposure to HMOs is the greatest. So, while there is sufficient justification to examine the association at this age, the long-term impact of HMOs during childhood, adolescence and adulthood is yet to be identified.

The current evidence for the role of HMOs in AD and FA prevention is inconsistent [85,86,89]. Only specific types of HMOs have been investigated. Although the major ones were mostly covered, these do not represent the diversity of HMOs. Some specific HMOs, in combination with specific gut bacteria and under specific environmental conditions, may be involved in AD or FA prevention. The HMOs present in breast milk collected and analysed only at one time point would not be representative of HMO composition over the whole lactation phase as some studies demonstrate that HMO composition changes over time [63,65]. Depending on the mother’s genetics, four milk groups have different HMO profiles and compositions. This is a potential confounding factor that should be accounted for when assessing the impact of HMOs on infant AD and FA. Hence, more longitudinal studies are required which collect and analyse breast milk at multiple time points to fully understand the dynamic changes of HMOs and its impact on infant health and well-being. Moreover, current studies did not consider immunological factors in breast milk. On average, mature milk contains approximately 10^8^–10^9^/L of leucocytes [90] but if not measured in each breast milk sample, its effects on allergy prevention cannot be controlled for.

At present, liquid chromatography is the most commonly used method to quantify HMOs in breast milk [62]. The technical difficulties are due to not only the small number of available standards but also the complexity and diversity of the structures [91]. Milk contains complex mixtures of biomolecules and HMOs first have to be isolated, removing liquids, lipids, salts and proteins. Out of more than 100 HMOs identified, only ~20 can be quantified and are used to estimate the total HMO concentrations [92]. This means that beneficial effects of HMOs are highly dependent on the variations in the concentrations of the uncharacterized HMOs. Finally, there is no standard analytical method used in all studies to compare HMO composition. Many studies have used the absolute concentration of HMOs, but the relative proportions of HMOs can also be used [92]. Depending upon whether absolute or proportions of HMOs are measured, the clinical outcome may vary.

## 6. Conclusions

The potential effects of HMOs on AD and FA are still speculative and require further investigation. More work needs to be done on the influences of components of breast milk on AD and FA, rather than the influence of breast milk itself, as components vary between individuals. Observational studies suggest an association between HMOs and AD and FA. However, underlying mechanisms of HMOs in reducing AD or FA outcomes are yet to be discovered. Large, longitudinal studies are required to better understand the impact of HMOs at various time points during breastfeeding and how long these impacts last after breastfeeding has ended. Exploring the short- and long- term impact of HMOs on AD and FA in infancy, adolescence and even adulthood is crucial and will provide evidence for other health benefits associated with breastfeeding.

## Figures and Tables

**Figure 1 nutrients-12-03212-f001:**
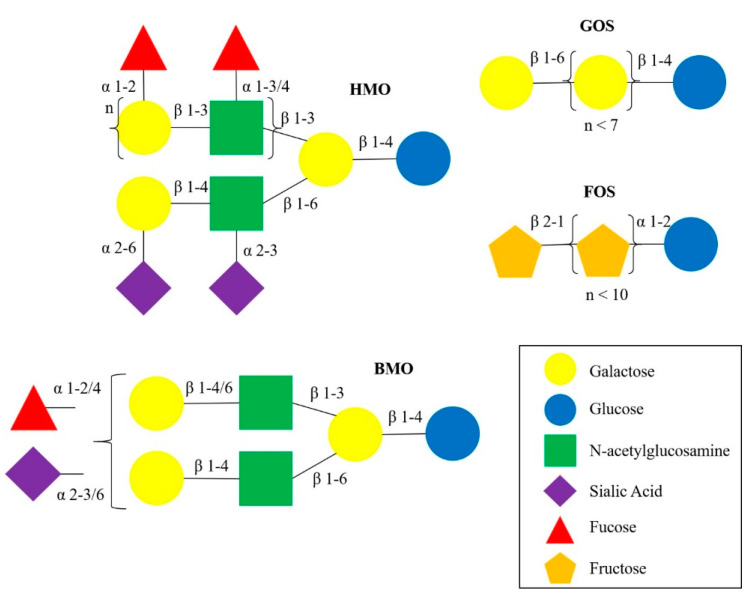
Structural schematics of HMO, BMO, GOS and FOS.The HMO structure is more complex and diverse compared to BMO and plat-derived oligosaccharides, GOS and FOS. HMO and BMO structures can be branched with various α- and β-linkages, GOS and FOS contain linear chains of repeating units of galactose (*n* < 7) and fructose (*n* < 10), respectively. Abbreviations: HMO, human milk oligosaccharide; BMO, bovine milk oligosaccharide; GOS, gallacto-oligosaccharide; FOS, fructo-oligosaccharide.

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
