# Peer review of "Do Human Milk Oligosaccharides Protect Against Infant Atopic Disorders and Food Allergy?"

_nutrients, 2020, doi:10.3390/nu12103212_

Round 1
Reviewer 1 Report
This is a nice commentary on the current landscape of human milk influences on the development of atopy/food allergy. The piece explores the previous epidemiological studies that link FA to milk consumption, with some interesting observations in clinical cohorts. The authors then discuss some of the components and potential mechanisms of action. Some potential areas for the authors to consider:
Major:
1) Discussion into human milk components in section 3 should include a discussion about probiotics inherently present in maternal milk, particularly if the mechanism mainly discussed in HMOs evolve around prebiotic mechanisms. Previous work by the Swedish group showed some work on potential TGFb response and Lactobacilli population that could be discussed here. PMID: 17349686, 18631345. (Prescott 2009, Abramhamsson 2008). The incidence, particularly amongst preterm infants, and the microbiota signatures that fluctuate in preterm infants with antibiotic usage could be discussed (PMID: 32888417).
2) Human milk oligosaccharides could use some diagrams to illustrate structural variation, and particularly draw out some of the differences between HMOs and plant-derived prebiotics.
3) Lines 126-136: the importance of lactotypes and HMO composition is a bit descriptive here, but if the point is about role in AD and FA, a discussion about whether the Lewis/Secretor subgroups have functional implications on FA/AD pathogenesis should be discussed. I.e. 1) are there differences in FA/AD rates amongst nations with higher secretors vs non-secretors and etc. 2) Similarly, what are the HMOs found in each subgroup? I think this could be a basic information for readers to appreciate the difference in composition, and then authors could discuss further about the purported mechanisms of some of those HMOs in each group.
4) Lines 178-182: this section could be expanded considering the immunomodulatory effects in clinical FA. There has been a wide selection of work in nutrition that show the different direct modulation, a discussion on the microbe-independent effects could be relevant - particularly given in early life the infant microbiota is not yet established to completely metabolize all forms of HMOs. Some studies to consdier PMID 30407734, PMID 31800974, PMC4183735.
Author Response
Thank you for your comments and suggestions. Please see attached for detailed responses.

Reviewer 2 Report
This manuscript is very interesting and analyzes carefully one very interesting problem analyzing the possible role of human oligosaccharides of the human milk in the possible prevention of the atopic dermatitis and food allergy in infants
The authors have analyzed the different situations, the differences found in breastfeeding and non-breastfeeding infants and the influence of prevention and the moment of onset of AD and FA
The mechanisms of protection of the HMOs have analyzed and the need to perform prospective and longitudinal studies are remarked.
Q1 : Title
Is appropriate and also is correct to write in an interrogative form
Q2 : Abstract
Line 17 : Please change data is, for data are
Line 22 : Please change preventative, for preventive
Q3 : 1. Atopic Disorders and Food Allergy in Childhood
Line 50 : Please specify the country where was realized this study (Ref. 10)
Line 62 : Please change antihistamines, for anti-histaminics
Q3 : 2 . Does Breastfeeding Modulate AD and FA Risks?.
This is a good exposition of the conflicting effects of the length of breastfeeding duration in the prevention of AD, eczema and asthma and FA in children and the distinct influence in adolescents and adults also
Q3 : 3. What is the role of human milk components in AD and FA prevention?
Very well explanation about the role of HMO content and the variability in the colostrum in their composition
Q3.4 . Could HMOs be related to AD and FA prevention ?
Line 166 : Please change abundance for presence
The influence of breastfeeding on the intestinal microbioma, through the presence of HMOs, is clearly exposed
Nevertheless there is no a clear influence of HMOs in the long-term impact on AD and/or FA in younger people and adults
Q3.5. What are the limitations to overcome?
Only specific types of HMOs have been investigated and the methods of quantification are not standardized
Q4. 6. Conclusions
This field of research is clearly open to discussion and is necessary to perform more longitudinal studies in order to clarify their effects bot at short term in the infancy and also at long-term in adulthood
Q,5 . References
Are good enough and well selected
Author Response
Thank you for your feedback. Please see the attachment for detailed responses.

Reviewer 3 Report
The manuscript is a commentary questioning the role of human milk oligosaccharides(HMO) in preventing infant atopic disease and food allergy. The authors did an excellent job reviewing of the literature and presenting the current status of HMO related to prevention of atopic disease and food allergy. They also provided future directions for research in this area. This will be interesting to the readers.
Author Response
Thank you for your positive feedback. We are glad to hear you liked our manuscript.
Reviewer 4 Report
Review of “Do Human Milk Oligosaccharides Protect Against Infant Atopic Disorders and Food Allergy?” (Soo Min Han et al, submitted 28 September 2020).
Summary: This review addresses the role for human milk oligosaccharides (HMO) in preventing allergic diseases including atopic dermatitis, food allergy, and asthma. It briefly defines the scope of the problem of atopic disease, reviews the role of breastfeeding in modulating atopic disease risk (highlighting some of the contradictory findings in the literature), reviews the components that comprise human milk, and then discusses the possible for the HMOs in preventing atopic disease. It also addresses limitations in how HMOs are identified and quantified in human breast milk that have slowed our ability to study their impacts on atopic disease.
General Comments: The review is well written and covers an important topic that merges the fields of nutrition and atopic disease. The last section especially does a good job of presenting the challenges to understanding / studying whether or not HMOs can impact atopic disease, highlighting aspects for researchers to tackle in future studies. There are a couple of sections within the review, particularly in the “Does Breastfeeding Modulate AD and FA risks?” where the review would benefit from the authors adding a sentence or 2 to try to reconcile or explain contradictory findings within the literature when it comes to breastfeeding’s impact of atopic diseases. Otherwise, I think this is a useful contribution to the literature on HMOs and atopic diseases.
Specific Comments:
- Line 44: 20% of children (or 10% of children) across the globe? Across the US? Across Europe? This needs to be specified.
- Line 61: change “persistence” to “Management”
- Line 74-75: Specify the study or studies that showed that the protective effect on eczema with exclusive breastfeeding over partial breastfeeding by author name (So and so et al showed…) It will make this section easier to understand.
- Line 78: Again, here you should specify which authors -"By contrast, (The authors of ref 32) showed that in the first 7 years, the risk of eczema increased...."
- Line 79: At the end of this sentence, you add a sentence or 2 speculating as to why these different studies showed contrasting findings. What is your perspective on the contrast between the two studies?
- Line 86-88: Again, can the authors add a sentence or 2 trying to reconcile these contradictory findings? Why do they think that findings vary so much between ref 35, 36, and 30? Were different definitions used for asthma? Were there different asthma phenotypes within a cohort? Were the studies done on populations with similar or different environmental circumstances, socioeconomic backgrounds, self-identified race and ethnicity? (For example).
- Line 167-168: At the end of this sentence, the authors should try to tie this ending sentences back toward HMO by adding a sentence after this one saying something like: "Additional studies exploring links between HMOs, early gut microbiome development and atopic susceptibility are required."
- Line 178-182: This paragraph should be rearranged as follows: “Lastly, HMOs can directly modulate the immune system. It was shown in vitro that HMOs reduce the production of pro-inflammatory cytokines and increase the levels of cytokines involved in tissue repair and homeostasis [72]. It is proposed that 1% of HMOs are absorbed into the systemic circulation [56]. This suggests that in vivo, HMOs may suppress the inflammatory reaction associated with atopy.”
- Line 189-190: Replace “could not be” with “were not” and “could be” with “were”
- Line 191: “lower risk” of what? “Lower risk” should be changed to “lower risk of allergic sensitization to food was associated with…”
- Line 194-195: It is uncertain if HMOs have long-term impact on what? Please clarify.
Author Response
Thank you for your comments. Please see the attachment for detailed responses.
